# Involvement of Vitamin D3 in the Aging Process According to Sex

**Daniela Florina Trifan** [1,†], **Adrian Gheorghe Tirla** [2], **Calin Mos** [2], **Adrian Danciu** [1], **Florian Bodog** [2], **Felicia Manole** [2,†] **and Timea Claudia Ghitea** [3,*,†]

1   Doctoral School of Biological and Biomedical Sciences, University of Oradea, 410087 Oradea, Romania; trifan.daniela17@yahoo.com (D.F.T.); adrian.danciu@telios.ro (A.D.)
2   Faculty of Medicine and Pharmacy, Medicine Department, University of Oradea, 410068 Oradea, Romania; adriantirla@gmail.com (A.G.T.); drmoscalin@yahoo.com (C.M.); fbodog@gmail.com (F.B.); fmanole@uoradea.ro (F.M.)
3   Faculty of Medicine and Pharmacy, Department of Pharmacy, University of Oradea, 10, 410068 Oradea, Romania
*   Correspondence: timea.ghitea@csud.uoradea.ro
†   These authors contributed equally to this work.

**Abstract:** Background/Aim: Rapid onset of facial ptosis can impact physical appearance and compromise the outcomes of facelift procedures. The level of vitamin D has a potential correlation with collagen formation and its deficiency with inflammatory processes that affect the breakdown of hyaluronic acid. This study aims to investigate the potential relationship between accentuated facial ptosis in women and low levels of vitamin D. Furthermore, it aims to explore preventive measures or strategies to slow down facial ptosis and enhance the longevity of facelift results. Materials and Methods: The study was focused on monitoring the vitamin D levels in women and men with advanced facial ptosis and comparing them with a control group. Results: Notably, a direct association between gender and serum vitamin D levels was observed, indicating less sustainable outcomes in women. Conclusions: Women face additional challenges in the aging process due to hormonal shifts after menopause or premenopausal, which are associated with osteoporosis and lower vitamin D levels.

**Keywords:** face-lifting; vitamin D; collagen; sex difference





## 1. Introduction

The phenomenon of aging is an inevitable, continuous, slow, dynamic natural physiological process [1,2]. The alteration of aesthetic appearance plays an important role in the development of social isolation and the onset of depression, and affects self-esteem [3]. As providers of aesthetic services, it is important to understand the improvement in quality of life that can be achieved through various aesthetic procedures, best measured by patient-reported outcomes. Several factors contribute to this growing trend of facelift interventions.; patients are not only seeking to enhance their physical appearance but also desire to improve their mental and emotional health, along with social well-being. In an interview-based study on the motivations of patients considering cosmetic dermatological procedures, concerns related to mental and emotional health were mentioned more frequently than any other identified themes, citing the desire to enhance confidence, feelings of depression and anxiety regarding their current appearance, and the burden of efforts to conceal undesired physical features as motivations for pursuing cosmetic procedures [4].

As one ages, the skin undergoes a series of biochemical changes. Cellular regeneration occurs at a slower rate, leading to loss of elasticity and dehydration of the skin [5]. A reduction in collagen levels leads to a decline in skin elasticity [6], as well as muscle mass loss beyond just the facial area [7]. Additionally, the degradation of collagen and elastin fibers contributes to the growing visibility of wrinkles. Part of the process of chronological

aging is genetically, metabolically, and hormonally determined, with skin deterioration also a result of exposure to ultraviolet radiation, gravitational force pulling the skin downward, vices, lack of sleep, unhealthy dietary habits, and stress. In the third to fourth decades of life, brow descent with soft tissue occurs, along with the descent and atrophy of the buccal fat pad, the appearance of fine lines around the eyes, deepening of expression lines at the corners of the mouth, nasolabial folds and marionette lines. Facial bone resorption occurring in the sixth to seventh decades of life and beyond further contribute to changes in appearance. Not all patients exhibit uniform deflation or uniform fullness of the face. Some patients present a mixed, asymmetric image, with fullness in one area, or a more pronounced ptosis of one side of the face with more significant excess skin in the zygomatic region and the mandibular region, or after cranioplasty for different diseases that affect the face [8].

Low vitamin D levels have been associated with reduced bone mineral density [9]. This was been the primary known effect until recently. However, in recent decades, an increasing number of studies have focused on the effects of vitamin D, both in the inflammatory response at the intestinal level [10,11] and in sarcopenia [12–14]. At the skin level, there is a correlation between vitamin D and the fortification of skin structure, as vitamin D stimulates the secretion of collagen [15]. Vitamin D is considered the new anti-inflammatory agent [16–18].

Vitamin D levels vary according to sex has been studied recently, but with very contradictory results [19–22]. It was found that women have a higher prevalence of hypovitaminosis than men after the age of 54 years, but the serum levels of vitamin D were not significantly different [19].

At the skin level, vitamin D has been correlated with reduced expression of collagen I and III, and other collagen isoforms [23]. A visible reduction in signs of aging has been observed following regular consumption of collagen [24], this is directly correlated with a high level of vitamin D [25], and deficiency of vitamin D receptor in keratinocytes augments dermal fibrosis and inflammation [26]. Articular cartilage erosions are correlated with low levels of vitamin D; thus it can be concluded that collagen formation is vitamin D dose-dependent.

The aim of this study was to establish whether there is a connection between facial lifting in both men and women and the serum level of vitamin D. This study intended to find the answer as to why the structure of facial skin is less firm in women and why women are keener to undergo lifting interventions. The primary objective of a facelift is to initiate a method that prevents or dampen the progress of facial ptosis, while also ensuring the maintenance of the achieved facelift results. Could pronounced facial ptosis in women be correlated with a low level of vitamin D? Could supplementation with vitamin D potentially dampen the progress of facial ptosis? In this study, our objective was to investigate potential variations in vitamin D levels between women and men who experienced advanced facial ptosis. Our aim was to determine whether sustainable results could be achieved in comparison to a control population with physiological facial ptosis.

## 2. Materials and Methods

A 12-month observational study was performed on 192 patients with facial ptosis who underwent aesthetic procedures, and a retrospective study was conducted on the correlation between cost and return visits following interventions of a group of 167 patients followed-up for 15 years. The patients presented themselves at the medical aesthetics office, where they were clinically and paraclinically evaluated. The patients were divided into two groups based on the type of intervention: 36 individuals underwent non-invasive facelifting with suspension threads, while 156 individuals underwent minimally invasive face-lifting procedures, either surgically or by hybrid method that includes both surgical intervention and resorbable threads. The patients visited the medical aesthetics office for clinical and paraclinical evaluations. To assess the quality of life, participants completed questionnaires at the beginning and end of the study period, with reference to both quality of life and body

dysmorphism [27–29]. The study excluded individuals receiving psychiatric treatment, those with coagulation disorders, severe heart conditions, or tumor conditions, pregnant women, those with facial paralysis, and those with inflammatory or infectious diseases in the face. Relative contraindications included autoimmune diseases and uncontrolled diabetes.

Non-invasive facelifting incorporated hyaluronic acid injections, Botox, and medical lifting techniques.

During surgical lifting, the subcutaneous musculo-aponeurotic system was folded using non-absorbable threads. Additionally, the lower portion of Bichat's Bula was anchored, and excess skin was excised. The minimally invasive facelift with Happy Lift—Anchorage Threads (Milano, Italy) was used, the thread thickness is 2-0 resorbable, with a size of $1 \times 31.6$ cm. Incorporating both surgical lifting and minimally invasive lifting techniques, the procedure involved the use of Happy Lift—Anchorage Threads. These threads, with a resorbable thickness of 2-0 and a size of $1 \times 31.6$ cm, provide an innovative approach that ensures enhanced long-term stability of the post-interventional results [30].

Return to the plastic surgery office is defined as the period during which the patients undergo some interventions. This period depends on the type of intervention, the gender of the person, but also the stage of aging.

Clinical and paraclinical investigation. The clinical assessment took place at the medical office (Estetic DYN Clinic, Oradea, Romania), while the evaluation of paraclinical parameters was conducted at authorized laboratories. The paraclinical examinations included checking the coagulogram, blood pressure, and blood sugar levels. Furthermore, a thorough local clinical examination was performed, taking into consideration the comprehensive medical history and factors such as smoking, alcohol consumption, and narcotic drug use.

Ultrasound examination. Shear-wave elastography, performed by a highly experienced radiologist, was utilized to assess the elasticity of the dermis and subcutaneous cellular tissue through ultrasound examinations. The examinations were conducted using an Aixplorer SuperSonic Imagine ultrasound machine (version 11.2.0) from Aix en Provence, France, with a linear probe (SL15-4) featuring a variable frequency range of 5–14 MHz. Measurements of shear-wave elasticity were taken at six locations on the face for each patient: three on the right half and three on the left.

During the examination, care was taken to avoid applying excessive pressure with the transducer, ensuring a noticeable gel layer was maintained between the transducer and the skin. The settings of the device were adjusted to display the elasticity values of the skin and subcutaneous cellular tissue in kilopascals. The surgical team determined the specific measurement sites, which included the zygomatic arch, Bichat's bubble, and lateral to the labial commissure. At each location, three measurements were taken in the dermis and three measurements were taken immediately beneath the skin in the subcutaneous cellular tissue. The device was configured to display the ratio between the values obtained at these two levels [31]. An initial ultrasound examination (Figure 1A) and a final ultrasound examination (Figure 1B) were conducted for each participant.

Rapid vitamin D test. The JusChek (Bucharest, Romania) rapid test was employed to conduct vitamin D testing and colorimetrically determine the baseline vitamin D level. As a result, there are four categories for interpreting the results: Insufficient $\leq$10–30 μg/mL; adequate: 30–99 μg/mL; optimal: 100 μg/mL; excessive >100 μg/mL. It was recommended to supplement with 2000 IU of vitamin D for a duration of 3 months, without making any nutritional interventions during this phase.

Before proceeding with the facelift, vitamin D testing was conducted. When the test results indicated an insufficient level of vitamin D, treatment involving vitamin D supplementation was recommended. After completing the treatment, another vitamin D test was performed. If the vitamin D levels were found to be sufficient, the decision was made to proceed with the facelifting procedure.

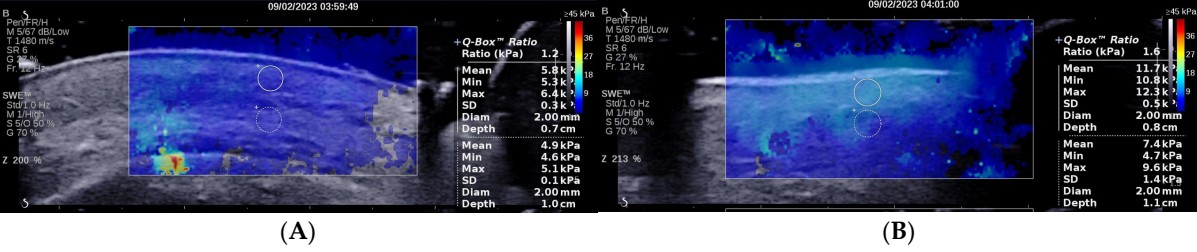

**Figure 1.** Prior to the minimally invasive intervention, elastography of the soft tissues was performed. (**A**): A sample with a size of 2 mm was utilized, initially at the dermis level and subsequently immediately below the dermis, adjacent to the original sample (**A**). (**B**): After 1 week, the final result for each location was determined by calculating the arithmetic mean of the three measurements taken at the dermis level, as well as the arithmetic mean of the three subdermal measurements. Additionally, the arithmetic mean of the ratios obtained from the three measurements was calculated.

To test the hypothesis quickly, that vitamin D levels may influence facelift outcome differently in women versus men, a rapid test was conducted.

Quality of life. Specific to the four different domains: physical health, psychological well-being, social relationships, and environment, were identified in the WHOQOL-BREF questionnaire assessing quality of life. The short questionnaire for the Big Five personality traits (BFI-10), which is the shortest personality questionnaire, was used, and symptoms of body dysmorphic disorder were evaluated using the Yale-Brown Obsessive-Compulsive Scale modified for Body Dysmorphic Disorder (Y-BOCS-BDD).

Statistical analysis. The data analysis was conducted using the Statistical Product and Service Solutions (version 20; IBM, Armonk, NY, USA) computer software program. Demographic variables, procedure frequency, and cost data, sourced from the medical office, were evaluated for the two surveyed time points and based on two study groups to identify significant trends. The means, frequency ranges, standard deviations, and tests of statistical significance were calculated using Student's *t*-test and chi-square test. The Bravais–Pearson correlation coefficient was utilized to measure the relationship between two variables. A significance level of $p < 0.05$ was considered statistically significant, while $p < 0.01$ indicated high-level statistical significance. Post-hoc analysis (Bonferroni) was employed for further subgroup analysis to analyze differences between groups.

## 3. Results

### 3.1. Demographic Data

Out of the 192 patients, 50 were male and 142 were female (*t* test, $t = 54.782$, $p = 0.001$), with ages ranging from 35 to 75 years (*t* test, $t = 65.852$, $p = 0.001$). The majority of the patients were from urban areas (83.85%) (*t* test, $t = 43.624$, $p = 0.001$). Among them, 36 individuals requested non-invasive interventions (such as hyaluronic acid filler injections, Botox, or microneedling), while 156 patients underwent various facelifting procedures (minimally invasive lifting with absorbable barbed sutures, traditional surgical lifting, combined surgical lifting), as presented in Table 1. It was observed that all women were from urban areas, as well as 38% of men, but all the men had higher education degrees (Anova test, $F = 30.057$, $p = 0.023$). Additionally, men were observed to have a significantly younger age ($F = 45.900$, $p = 0.001$), which can be explained by the smaller number of male participants in the study.

**Table 1.** Demographic description of study participants.

| Parameter | | Groups | | | |
| --- | --- | --- | --- | --- | --- |
| | | **Men** | | **Women** | |
| | | **Facelift** | **Control** | **Facelift** | **Control** |
| | | **N (%)** | **N (%)** | **N (%)** | **N (%)** |
| Origin | Urban | 16 (32.00) | 3 (6.00) | 123 (86.61) | 19 (13.38) |
| | Rural | 17 (34.00) | 14 (28.00) | 0 (0.0) | 0 (0.0) |
| | *p* | | 0.001 | | |
| Level of education | Secondary | 0 (0.0) | 0 (0.08) | 14 (7.3) | 1 (0.5) |
| | Higher | 33 (66.00) | 17 (34.00) | 109 (76.76) | 18 (12.67) |
| | *p* | | 0.001 | | |
| Age, years | Mean ± SD | 45.67 ± 5.10 | 43.53 ± 5.78 | 54.11 ± 11.95 | 53.42 ± 6.09 |
| | *p* | | 0.001 | | |
| Median (range) vitamin D level, μg/mL | Initial | 1.94 (0.24) | 0.76 (0.83) | 1.76 (0.43) | 0.89 (0.88) |
| | *p* | | 0.658 | | |
| Median (range) vitamin D level, μg/mL | Final | 1.97 (0.17) | 1.47 (0.62) | 1.73 (0.44) | 0.95 (0.62) |
| | *p* | | 0.023 | | |

N: Number of participants; SD: standard deviation; *p*: Statistically significance.

### 3.2. Level of Vitamin D

The initial-to-final differences in serum vitamin D levels were significantly different between males and females, as shown in Table 2, and also between the two groups ($p = 0.001$).

**Table 2.** Distribution of patients initial and final level of vitamin D.

| Time Point | | **Men** | | **Women** | |
| --- | --- | --- | --- | --- | --- |
| | | **Facelift** | **Control** | **Facelift** | **Control** |
| | **Vitamin D Level** | **N (%)** | **N (%)** | **N (%)** | **N (%)** |
| Initial | Insufficient: <10 μg/mL | 0 (0.0) | 8 (16.00) | 0 (0.0) | 8 (5.63) |
| | Sufficient: 30 μg/mL | 2 (4.00) | 5 (10.00) | 29 (20.42) | 5 (3.52) |
| | Optimal: 100 μg/mL | 31 (62.00) | 4 (8.00) | 94 (66.19) | 6 (4.22) |
| | Excess: >100 μg/mL | 0 (0.0) | 0 (0.0) | 0 (0.0) | 0 (0.0) |
| | *p* * | | 0.650 | | |
| Final | Insufficient: <10 μg/mL | 0 (0.0) | 1 (2.00) | 0 (0.0) | 4 (2.81) |
| | Sufficient: 30 μg/mL | 1 (2.00) | 7 (14.00) | 33 (23.23) | 12 (8.45) |
| | Optimal: 100 μg/mL | 32 (64.00) | 9 (18.00) | 90 (63.38) | 3 (2.11) |
| | Excess: >100 μg/mL | 0 (0.0) | 0 (0.0) | 0 (0.0) | 0 (0.0) |
| | *p* * | | 0.032 | | |

N: Number of participants. *p* *: Statistically significance Chi-Square.

Following data processing using the boxplot method (Figure 2A), it was observed that the differences in initial-to-final levels of serum vitamin D in the facelifting group did not differ significantly ($p = 0.394$). However, in the control group, significant differences were recorded ($p = 0.035$). Elastographic evaluations (Figure 2B) showed significant differences

in both the lifting and control groups between males and females. A greater difference was observed in males in the facelifting group, while in the control group, females exhibited a larger, statistically significant difference in both cases ($p < 0.05$).

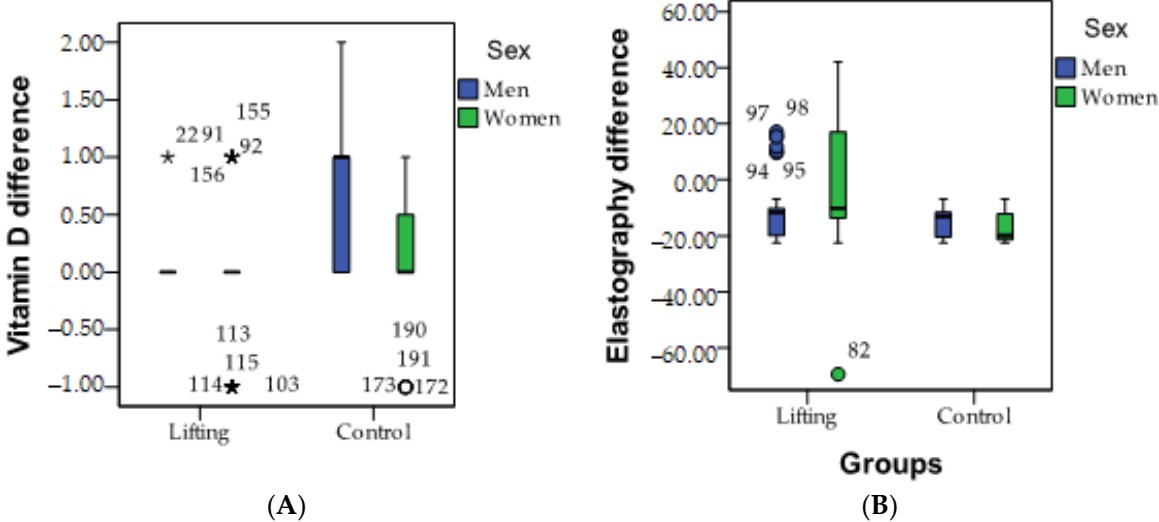

(**A**)  (**B**)

**Figure 2.** Boxplot of the difference in vitamin D in each study group according to sex (**A**), and the difference of elastography in each study group according to sex (**B**). In (**A**) the median line in the lifting group shows no statistically significant differences, whereas a significant difference can be observed in the control group. Patients 113, 114, 115, and 103 are exceptions, in which a decrease in vitamin D level was recorded, where "*" represents exceptional cases, and "°" less frequent cases. In (**B**), the standard deviation in the lifting group for women shows greater variation in elastographic differences than in men, with the average values being close. However, in the control group, the average value differs between men and women, but the smaller standard deviation indicates reduced variations.

### 3.3. Return to the Plastic Surgery Office

A group of 167 patients was followed-up for 15 years. The retrospective study on patient return following interventions did not show a significant difference between men and women within groups. For men in the lifting group, the average return time was $6.39 \pm 1.82$ years, and for men in the control group, it was $0.71 \pm 0.25$ years. Similarly, for women in the facelift group, the average return time was $6.28 \pm 3.04$ years, and for women in the control group, it was $0.66 \pm 0.24$ years. There were no statistically significant differences between women and men, but the 2 groups showed significant differences regarding the return to the medical office. The median line shows comparable values, but the first quartile for women shows a compact group, and the larger standard deviation indicates a greater variation, shown in Figure 3. Due to these variations, the conclusion was raised that the level of vitamin D contributes to increased variations. The return for touch-ups was on average 6 years, for both men and women, for men only some exceptional cases returned sooner or later, as can be seen from Figure 3, while for women the return was much longer frequency, as seen by the size of the standard deviation, in both the control and lifting groups, which may be explained by the fact that there were lower vitamin D levels for women in both the control and lifting groups and from the lifting group.

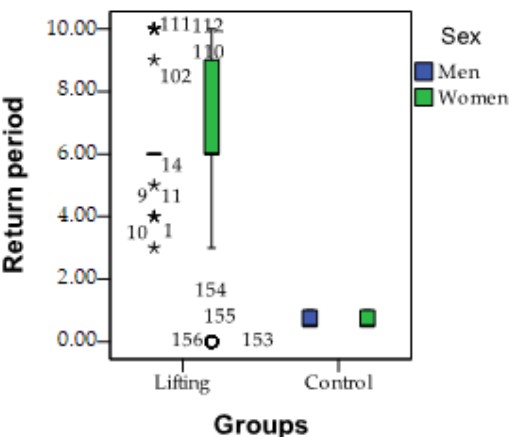

**Figure 3.** Graphic boxplot representation of the return period expressed in years, for each study group according to sex. Although the median line is at the same value, the lack of standard deviation in men shows that only exceptional cases returned faster ("*" represents exceptional cases, and "°" less frequent cases). On the other hand, in women, the high standard deviation indicates that not only exceptionally but significantly more often, women returned faster.

### 3.4. Cost

Regarding the costs of the interventions, a retrospective study was conducted on the correlation between cost and return. The evaluation was based on data adjustments, taking into account patients who did not return to the same clinic or opted for another procedure. It was observed that women had higher costs to achieve the same results as men due to undergoing more procedures, including both lifting and non-invasive interventions. In the lifting subgroup, the cost for men was recorded as 2186.36 ± 387.55 euros, while for women it was significantly higher at 2632.52 ± 820.31 euros ($p = 0.029$). In the control group, the cost for men was 647.06 ± 158.58 euros, and for women it was 700.00 ± 228.52 euros ($p > 0.05$), presented in Figure 4A. From the correlation between cost and return period, an $R^2$ value of 0.143 was obtained. As the cost increases, so does the return period, as shown in Figure 4B. That means lower cost means faster return.

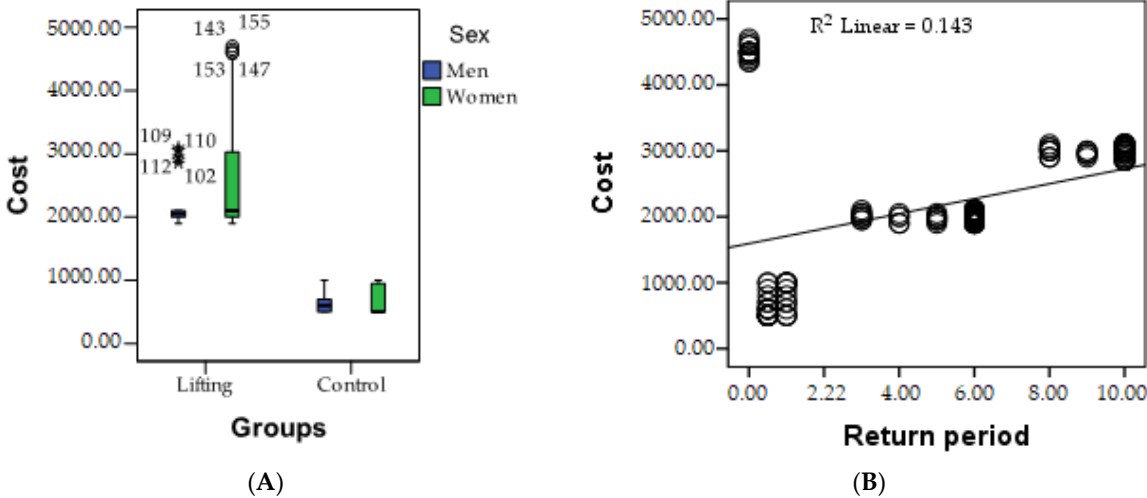

(**A**)                       (**B**)

**Figure 4.** Graphical representation of the costs of the interventions in euros (**A**), and the correlation between cost and return period in years (**B**). In (**A**), the identical median line indicates the same costs for interviews in men as in women, but the lack of standard deviation in men shows that only exceptional cases had different costs, where "*" represents exceptional cases, and "°" less frequent cases. In contrast, the large standard deviation in women shows that not only exceptionally but significantly more often, they had additional costs.

### 3.5. Risk Factors

For any cosmetic procedure, whether surgical or non-surgical, there are risk factors that can lead to reduced durability of the results and, in some cases, even jeopardize the patient's health. Therefore, monitoring risk factors is essential. We monitored hypertension, smoking, alcohol consumption, and the presence of type 2 diabetes mellitus, described in Table 3.

**Table 3.** Descriptive statistics of risk factors.

| Parameter | | Men | | Women | |
| --- | --- | --- | --- | --- | --- |
| | | Facelift | Control | Facelift | Control |
| | | N (%) | N (%) | N (%) | N (%) |
| HTN | No | 27 (54.00) | 14 (28.00) | 82 (57.74) | 13 (9.15) |
| | Yes | 6 (12.00) | 3 (6.00) | 41 (28.87) | 6 (4.22) |
| *p* | | | | 0.044 | |
| Alcohol consumption | No | 24 (48.00) | 10 (20.00) | 91 (64.08) | 14 (9.85) |
| | Yes | 9 (18.00) | 7 (14.00) | 32 (22.53) | 5 (3.52) |
| *p* | | | | 0.420 | |
| Smoking | No | 32 (64.00) | 14 (28.00) | 107 (75.35) | 17 (11.97) |
| | Yes | 1 (2.00) | 3 (6.00) | 16 (11.26) | 2 (1.40) |
| *p* | | | | 0.373 | |
| DM2 | No | 27 (54.00) | 14 (28.00) | 108 (76.05) | 16 (11.26) |
| | Yes | 6 (12.00) | 3 (6.00) | 15 (10.56) | 3 (2.11) |
| *p* | | | | 0.353 | |

DM2: Diabetes melitus type 2; HTN: hypertension; N: number of participants; *p*: Statistically significance.

A higher incidence of hypertension was recorded in women in the facelifting subgroup. This can be explained by the fact that the minimally invasive procedure allows intervention for individuals with hypertension, as it does not require general anesthesia. Smoking prevalence was higher among women in the facelifting subgroup, while the difference was statistically non-significant in the control group. There were no statistically significant differences between men and women in either the facelifting subgroup or the control group.

### 3.6. Quality of Life

It was observed that in the lifting group, the difference between WHOQOL-BREF scores was higher for men than for women, as was the case with BFI-10 scores, while the difference between Y-BOCS-BDD scores was $-7.70 \pm 7.20$ for men and $-9.49 \pm 7.47$ for women, indicating that men were more psychologically balanced. The same trend can be observed in the control group, but without statistically significant differences in both groups, as shown in Figure 5.

### 3.7. Risk Factor for Vitamin D Deficiency

The Spearman's rho correlation measures a monotonic relationship using ranked data between the two variables, namely the disparity in vitamin D levels and sex. Within the facelift group, there was a significant disparity in vitamin D levels, with women experiencing a more pronounced reduction compared to men ($p = 0.003$). In the male gender, the serum level of vitamin D shows a statistically significant increase both initially and at the end in the lifting group, with a more pronounced increase observed at the end ($p = 0.032$), presented in Figure 6. In the control group, initially, no significant differences were recorded between the sexes in the serum level of vitamin D ($p = 0.653$), but at the end, a significantly higher increase in the serum level of vitamin D was observed in men.

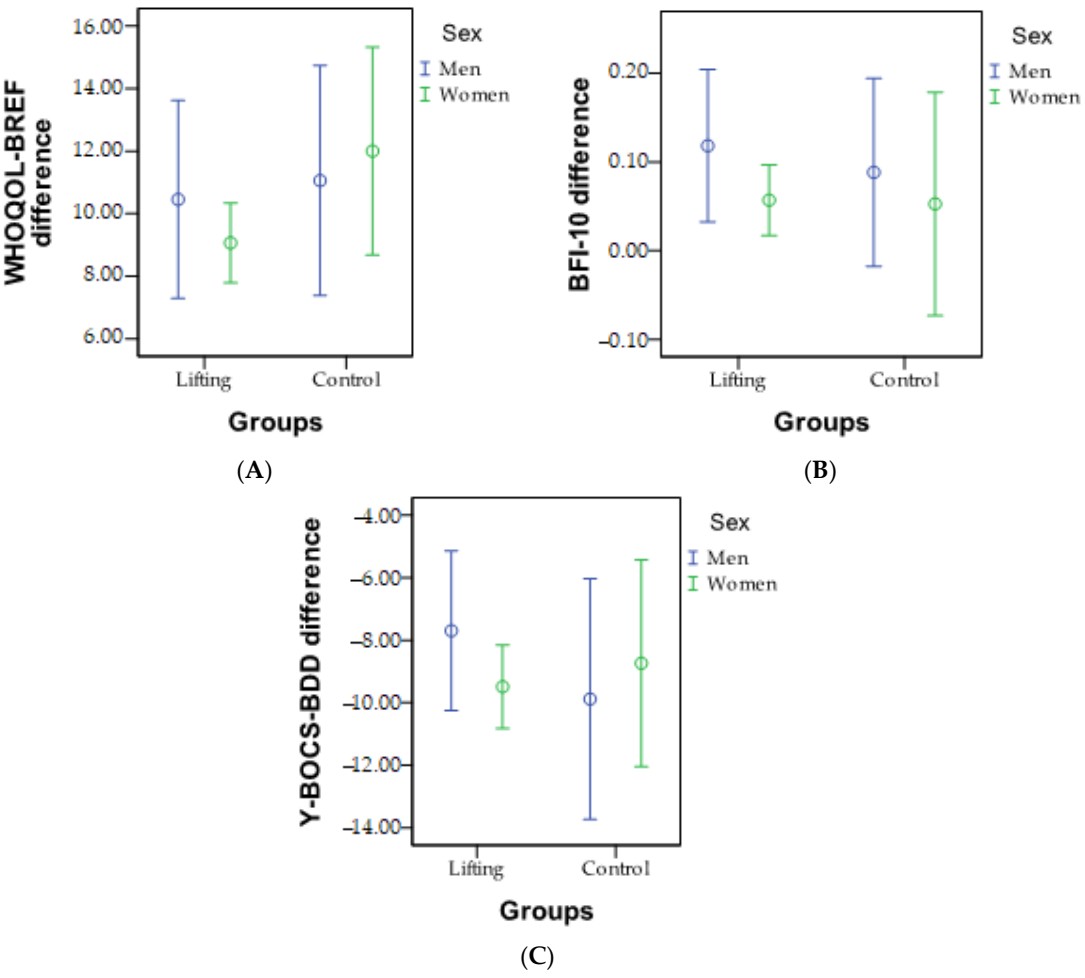

**Figure 5.** Mean of quality of life difference with 95% confidence interval for World Health Organization Quality of Life Brief Version (WHOQOL-BREF (**A**)), for Big Five personality traits (BFI-10 (**B**)), and for Yale-Brown Obsessive-Compulsive Scale (Y-BOCS-BDD (**C**)) in in each study group.

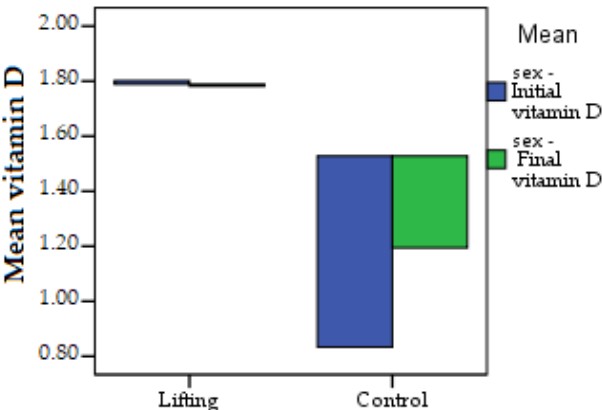

**Figure 6.** Graphical representation of the relationship between vitamin D and sex at initial and final within the two study groups.

## 4. Discussion

Facial lifting interventions have long been seen as primarily benefiting women. However, over time, men have also embraced the idea of facial lifting procedures, recognizing the potential to boost their self-confidence and improve their quality of life. In general, there is a strong correlation between educational level and facial lifting procedures in men,

as supported by previous studies [27,28], which aligns with the findings of our study. For women, educational level is also important but a lower educational level does not preclude the possibility of undergoing facelifting interventions.

Vitamin D deficiency has been extensively discussed in the past decade. A study reported a significant deficiency of vitamin D in Turkey [29], which affected women more than men. Another study examined vitamin D levels, where its changes were found to be dependent on various factors, including sex [30]. Considering scientific evidence that indicates different synthesis and metabolism of vitamin D in women and men, it has been concluded that the results are contradictory [21], and besides the patient's sex, several other factors come into play as obesity or cardiometabolic diseases [20]. In our study, we observed lower vitamin D levels in women compared to men, and despite having identical supplementation recommendations for both sexes, we were unable to achieve comparable serum vitamin D levels, with women exhibiting lower levels. Gender disparities in gut microbiota have been observed across various animal and human studies by Kim in 2020 [31]. This can be explained by the close relationship between the microbiome in intestinal homeostasis, respiratory tract function, and other organs, and the level of vitamin D [32].

Vitamin D is involved in collagen synthesis [24,25,33] and its deficiency is correlated with profibrotic factors by inducing an antifibrotic phenotype in multipotent mesenchymal cells. Studies have been conducted to test vitamin D supplementation as a preventive or early treatment strategy for cardiovascular diseases and associated fibrotic disorders [23,26] and concluded that vitamin D decreases fibrosis. In the current study, the return time for touch-up and reapplication was faster in women compared to men. This can be attributed to reduced collagen synthesis in women due to lower serum vitamin D levels [34].

Facelifting procedures have been performed on men for several decades. Since the 70s, SMAS facelifts have been applied for the first time to both men and women [35]. However, it is known that male facial lifts gained popularity in the mid-20th century, with advancements in surgical techniques and increasing acceptance of cosmetic procedures for men [36]. Due to the predominantly female nature of facelift procedures, techniques have evolved primarily to address facial aging in women. Techniques have emerged that allow for facial rejuvenation in men while maintaining a natural, masculine appearance devoid of surgical signs [36]. This can explain why less surgical intervention is needed for men compared to women in the current study.

Risk factors in facial plastic surgery are increasingly under control. While they were once a cause for concern in the first half of the previous decade, modern minimally invasive techniques [5] have enabled the monitoring of these risk factors without considering them as exclusion criteria. In our study, incidence rates of hypertension and smoking were higher in women, while men had a higher incidence of alcohol consumption, although these differences were statistically insignificant ($p > 0.05$).

Quality of life is measured through observers' perceptions of facelifting surgery and has been associated with opinions that, postoperatively, patients appear younger, more attractive, healthier, and more successful [37–40]. If until 2018, risk factors (related to surgical interventions) significantly influenced the results of facelifts in men [41], after this year, thanks to modern minimally invasive techniques, these risk factors became insignificant, as evidenced by the current study.

Deficiency of vitamin D receptor in the epidermis is associated with increased dermal thickness [42], infiltration of inflammatory cells [43], and severe collagen deposition [26]. An increase in the number of vitamin D receptors leads to collagen overproduction (COL1A1, COL1A2, COL3A1) and pro-inflammatory cytokines as interleukins (IL-1β, IL-6) and chemokines (CXCL1, CXCL2) [5,26,44]. Can be explained by hormonal expression and differences in bone mass between sexes [10,45], which contribute to different homeostasis mechanisms depending on the sex [46].

The aim of the study is to investigate whether vitamin D supplementation in women can reduce the risk of accelerated aging. This could be a future question regarding the long-term improvement and maintenance of facelifting results in both women and men.

## 5. Conclusions

Men have higher serum levels of vitamin D compared to women with 9.44% at the final of study. The time it takes for men to return to the aesthetic surgery office for reapplication, filling, and touch-ups is longer than that of women. Among the four monitored risk factors, three had a higher occurrence in women. This higher incidence in women can be attributed to the fact that they undergo lifting interventions more frequently than men, but it may also have other causes.

There is a strong correlation between vitamin D level and sex. This might explain why the outcomes of facelift procedures in women are less durable, costlier, and result in shorter return times. These findings are susceptible to alteration, should longer-term studies produce alternative data. Long-term studies with larger patient groups followed over an extended period might verify this hypothesis.

**Author Contributions:** Conceptualization, T.C.G.; methodology, D.F.T.; software, T.C.G.; validation, T.C.G. and D.F.T.; formal analysis, T.C.G.; investigation, T.C.G.; resources, A.D.; data curation, T.C.G.; writing—original draft preparation, T.C.G.; writing—review and editing, T.C.G.; visualization, C.M.; supervision, A.G.T.; project administration, F.B.; funding acquisition, F.M. All authors have read and agreed to the published version of the manuscript.

**Funding:** The APC was funded by University of Oradea, Oradea, Romania.

**Institutional Review Board Statement:** The study was conducted in accordance with the Declaration of Helsinki and approved by the Institutional Review Board (or Ethics Committee) of the University of Oradea (protocol code CEFMF/1 from 31 January 2023 and date of approval).

**Informed Consent Statement:** Informed consent was obtained from all subjects involved in the study. Written informed consent has been obtained from the patient(s) to publish this paper.

**Data Availability Statement:** All the data processed in this article are part of the research for a doctoral thesis, being archived in the aesthetic medical office, where the interventions were performed.

**Acknowledgments:** The authors would like to thank to the University of Oradea, for supporting the payment of the invoice, through an internal project.

**Conflicts of Interest:** The authors declare no conflict of interest.

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
