# Peer review of "Involvement of Vitamin D3 in the Aging Process According to Sex"

_cosmetics, doi:10.3390/cosmetics10040114_

Round 1
Reviewer 1 Report
Thank you for the opportunity to review the article titled "The Indirect Effects of Vitamin D3 on the Aging Process According to Gender." The study presents a compelling investigation into the potential relationship between vitamin D levels and facial ptosis, with a specific focus on women. The authors' exploration of the impact of vitamin D on collagen formation and its connection to inflammatory processes affecting hyaluronic acid breakdown is both timely and relevant, considering the significance
The study's methodology, which involved monitoring vitamin D levels in women and men with advanced facial ptosis and comparing them with a control group, provides a structured approach to analyze the subject matter. By incorporating gender-specific analyses, the study takes into account potential variations in aging processes between men and women, yielding valuable insights into the observed differences in serum vitamin D levels and their implications on sustainable outcomes.
One of the key strengths of the article lies in its acknowledgment of the broader challenges that women face during the aging process, particularly in relation to hormonal shifts post-menopause or premenopausal stages. The consideration of osteoporosis and lower vitamin D levels in this context adds depth to the study's findings and offers a comprehensive perspective on gender-related aging disparities.
Study limitations are the relatively small sample size and potential demographic homogeneity might limit the generalizability of the results. As a result, caution should be exercised when extrapolating these findings to broader populations. Additionally, the study's observational nature might hinder establishing a definitive causal link between vitamin D levels and facial ptosis. Despite these limitations, "The Indirect Effects of Vitamin D3 on the Aging Process According to Gender" contributes valuable information to the fields of aging research and cosmetic medicine.
Author Response
Reviewer 1
Thank you for the opportunity to review the article titled "The Indirect Effects of Vitamin D3 on the Aging Process According to Gender." The study presents a compelling investigation into the potential relationship between vitamin D levels and facial ptosis, with a specific focus on women. The authors' exploration of the impact of vitamin D on collagen formation and its connection to inflammatory processes affecting hyaluronic acid breakdown is both timely and relevant, considering the significance
The study's methodology, which involved monitoring vitamin D levels in women and men with advanced facial ptosis and comparing them with a control group, provides a structured approach to analyze the subject matter. By incorporating gender-specific analyses, the study takes into account potential variations in aging processes between men and women, yielding valuable insights into the observed differences in serum vitamin D levels and their implications on sustainable outcomes.
One of the key strengths of the article lies in its acknowledgment of the broader challenges that women face during the aging process, particularly in relation to hormonal shifts post-menopause or premenopausal stages. The consideration of osteoporosis and lower vitamin D levels in this context adds depth to the study's findings and offers a comprehensive perspective on gender-related aging disparities.
Study limitations are the relatively small sample size and potential demographic homogeneity might limit the generalizability of the results. As a result, caution should be exercised when extrapolating these findings to broader populations. Additionally, the study's observational nature might hinder establishing a definitive causal link between vitamin D levels and facial ptosis. Despite these limitations, "The Indirect Effects of Vitamin D3 on the Aging Process According to Gender" contributes valuable information to the fields of aging research and cosmetic medicine.
Response: Thank you for your kind words! We have re-evaluated the findings, reported on the study's conclusion (lines 373-374), and adjusted the title to prevent any reader misinterpretation. This study is intended to serve as a foundation for future research in this area. As authors of this manuscript, we express our gratitude for your insightful comments that have significantly enhanced the paper's quality. Your dedicated time and effort for this task are greatly appreciated.

Reviewer 2 Report
The manuscript titled: "The indirect effects of vitamin D3 on the aging process according to gender". By Timea Claudia Ghitea.
In this study, Timea Claudia Ghitea investigates the potential relationship between accentuated facial ptosis in women and low levels of vitamin D; monitoring the vitamin D levels in women and men with advanced facial ptosis and comparing them with a control group (12-month observational study was performed). This reviewer has the following comments:
Title: The title is misleading. The author indicates “The indirect effects of vitamin D3”. However, as far as this reviewer can tell, the author never identified an indirect effect of vitamin D.
Line 173: (p=0.001), I recommend indicating that statistical test and if what they compared was the percentage of participants by gender.
Line 174. (p=0.182), I recommend indicating that statistical test; since it is not clear if the age difference was grouped by gender.
Line 175. (p=0.001), I recommend indicating that statistical test
Line 176. It is mentioned that there are 158 patients, but the sum in Table 1 is 156
Line 179. It is mentioned that 62% of men live in urban areas, but 19/50 is 38%
Line 180. Men had higher education degrees (p=0.023), I recommend indicating that statistical test; since it is not clear if the difference in the level of education was grouped by gender.
Line 181. Men were observed to have a 180 significantly younger age (p<0.05), I recommend indicating that statistical test; since it is not clear if the age difference was grouped by gender.
Lines 186-187: Initially, a higher level of vitamin D was observed in the male control group compared to females (p=0.001). However, the median for the control group of men is 0.76 (0.83) lower than that for women is 0.89 (0.88). I recommend that you indicate which statistical test was used.
Lines 186-190: These lines don’t make sense with Table 1.
Lines 210-214: It is mentioned that it is a retrospective study and in materials that it is followed up for 12 months.
Lines 259-260: These lines don’t make sense with Table 3 (the percentages don’t correspond to what is observed in Table 3).
Finally, the author concluded,” There is a strong correlation between vitamin D level and sex. This might explain why the outcomes of facelift procedures in women are less durable, costlier, and result in shorter return times”. This is overestimated, since in this study the patient's follow-up was only done for 12 months.
Author Response
Reviewer 2
Firstly, I, the author of the present manuscript wishes to thank you for thoughtful commentary you have provided to improve the quality of the paper. I am very grateful for the time and effort you have devoted to this task. We have extensively revised my manuscript according to the recommendations. All changes in the text and the new figures that we have redesigned are highlighted. Please, see the point-by-point answers to your comments below. All correction was highlighted in the manuscript.
In this study, Timea Claudia Ghitea investigates the potential relationship between accentuated facial ptosis in women and low levels of vitamin D; monitoring the vitamin D levels in women and men with advanced facial ptosis and comparing them with a control group (12-month observational study was performed). This reviewer has the following comments:
Comment 1. Title: The title is misleading. The author indicates “The indirect effects of vitamin D3”. However, as far as this reviewer can tell, the author never identified an indirect effect of vitamin D.
Answer 1. Thank you very much for the comment. I have made corrections to ensure there is no potential for misleading information. „Involvement of vitamin D3 in the aging process according to sex”
Comment 2. Line 173: (p=0.001), I recommend indicating that statistical test and if what they compared was the percentage of participants by gender.
Answer 2. Thank you very much for the observation, the manuscript was completed accordingly (line 179).
Comment 3. Line 174. (p=0.182), I recommend indicating that statistical test; since it is not clear if the age difference was grouped by gender.
Answer 3. Thank you very much for the observation, the manuscript was completed accordingly (line 180).
Comment 4. Line 175. (p=0.001), I recommend indicating that statistical test
Answer 4. Thank you very much for the observation, the manuscript was completed accordingly (line 181).
Comment 5. Line 176. It is mentioned that there are 158 patients, but the sum in Table 1 is 156
Answer 5. Thank you for the amendment. It was a mistake. Please, see the correction highlighted in the manuscript (line 183).
Comment 6. Line 179. It is mentioned that 62% of men live in urban areas, but 19/50 is 38%
Answer 6. Thank you for the amendment. I corrected the manuscript. Please, see the correction highlighted in the manuscript (line 186).
Comment 7. Line 180. Men had higher education degrees (p=0.023), I recommend indicating that statistical test; since it is not clear if the difference in the level of education was grouped by gender.
Answer 7. Thank you very much for the observation, the manuscript was completed accordingly with test Anova (lines 186-187).
Comment 8. Line 181. Men were observed to have a 180 significantly younger age (p<0.05), I recommend indicating that statistical test; since it is not clear if the age difference was grouped by gender.
Answer 8. Thank you very much for the observation, the manuscript was completed accordingly with test Anova (line 188).
Comment 9. Lines 186-187: Initially, a higher level of vitamin D was observed in the male control group compared to females (p=0.001). However, the median for the control group of men is 0.76 (0.83) lower than that for women is 0.89 (0.88). I recommend that you indicate which statistical test was used.
Lines 186-190: These lines don’t make sense with Table 1.
Answer 9. Thank you for the amendment. The sentences were removed (lines 193-197)
Comment 10. Lines 210-214: It is mentioned that it is a retrospective study and in materials that it is followed up for 12 months.
Answer 10. Apologies for any confusion. This study constitutes an observational investigation spanning 12 months, focusing on the correlation with vitamin D. The analysis of return visits to the doctor's office was retrospectively carried out utilizing data extracted from the doctor's office registry. The methodology has been thoroughly understood. Please refer to lines 91-92 for more details.
Comment 11. Lines 259-260: These lines don’t make sense with Table 3 (the percentages don’t correspond to what is observed in Table 3).
Answer 11. Thank you for the amendment. The sentences were removed (lines 266-267)
Comment 12. Finally, the author concluded,” There is a strong correlation between vitamin D level and sex. This might explain why the outcomes of facelift procedures in women are less durable, costlier, and result in shorter return times”. This is overestimated, since in this study the patient's follow-up was only done for 12 months.
Answer 12. Thank you for your observation. This does indeed represent a limitation of the study, which I have included in the limitations section. The sentence I inserted reads: "These findings are subject to change if longer-term studies yield different data." Please see lines 373-374.

Round 2
Reviewer 2 Report
There is a need to elaborate on the methodology.
Profiling of the subjects (control and those with Facelift) must be well discussed in the methodology. There are also inconsistent (or at least confusing)
Line 95. “34 individuals underwent non-invasive facelifting with suspension 95 threads”.
Comment: in Table 1. Demographic description of study participants. the sum of controls is 36.
Lines 90-92. Regarding comment 10,
Comment: I suggest that you indicate in the material and methods:
“A 12-month observational study was performed on 192 patients with facial ptosis who underwent aesthetic procedures, and a retrospective study was conducted on the correlation between cost and return visits following interventions of a group of 167 patients followed-up for 15 years.
Remove the sentence (lines 346-347): This means that these risk factors 346 do not influence the results in a specific female or male gender.
Line 241: “A group of 167 patients was followed-up for 15 years”.
Comment: Move to line 217 which begins by describing retrospective study results and includes the number of women and men in this group.
Line 285-287: “The Pearson correlation reveals a robust association between the two variables, namely the disparity in vitamin D levels and sex. For the regression correlation calculation, we assigned the value "1" to males and "2" to females”.
Comment: Gender, which is a categorical variable (male and female). It is not correct to use the Pearson correlation coefficient, this is the test statistic that measures the statistical relationship, or association, between two continuous variables.
Author Response
Reviewer 2, Round 2
In the initial instance, we, the authors of the current manuscript, would like to express my gratitude for the considerate commentary you have furnished with the intention of enhancing the caliber of the paper. Your commitment of time and energy to this endeavor is deeply appreciated. Subsequent to your recommendations, my manuscript has been subjected to comprehensive revisions. All modifications made to the text, along with the introduction of new figures that have been restructured, have been distinctly marked. Kindly peruse the responses provided in a detailed manner to your remarks below. The entirety of the corrections has been conspicuously indicated within the manuscript.
There is a need to elaborate on the methodology.
Profiling of the subjects (control and those with Facelift) must be well discussed in the methodology. There are also inconsistent (or at least confusing)
Line 95. “34 individuals underwent non-invasive facelifting with suspension 95 threads”.
Comment: in Table 1. Demographic description of study participants. the sum of controls is 36.
Answer 1: Thank you for the thorough and highly intriguing observation. We have rectified the erroneous segments.
Lines 90-92. Regarding comment 10,
Comment: I suggest that you indicate in the material and methods:
“A 12-month observational study was performed on 192 patients with facial ptosis who underwent aesthetic procedures, and a retrospective study was conducted on the correlation between cost and return visits following interventions of a group of 167 patients followed-up for 15 years.
Answer 2. Thank you for the suggestion. We have made changes to the paragraph.
Remove the sentence (lines 346-347): This means that these risk factors 346 do not influence the results in a specific female or male gender.
Answer 3. Thank you for amendment, we removed the sentence.
Line 241: “A group of 167 patients was followed-up for 15 years”.
Comment: Move to line 217 which begins by describing retrospective study results and includes the number of women and men in this group.
Answer 4. Thank you for suggestion. We have moved the sentence in line 220.
Line 285-287: “The Pearson correlation reveals a robust association between the two variables, namely the disparity in vitamin D levels and sex. For the regression correlation calculation, we assigned the value "1" to males and "2" to females”.
Comment: Gender, which is a categorical variable (male and female). It is not correct to use the Pearson correlation coefficient, this is the test statistic that measures the statistical relationship, or association, between two continuous variables.
Answer 5. Thank you for observation. To ensure the most accurate representation of the results, we completely rewrote the regression section, opting for the Spearman correlation.
Thank you once more for your precious time and effort dedicated to the correction of our manuscript.

Round 3
Reviewer 2 Report
The manuscript has improved considerably.